# Automation of Property Acquisition of Single Track Depositions Manufactured through Direct Energy Deposition

**Jorge Gil** [1,2,*] ，**Abílio de Jesus** [1,2] ，**Maria Beatriz Silva** [3] ，**Maria F. Vaz** [3] ，**Ana Reis** [1,2]
**and João Manuel R. S. Tavares** [1,2]

1 Instituto de Ciência e Inovação em Engenharia Mecânica e Engenharia Industrial, Campus da FEUP, R. Dr. Roberto Frias 400, 4200-465 Porto, Portugal; ajesus@fe.up.pt (A.d.J.); areis@inegi.up.pt (A.R.); tavares@fe.up.pt (J.M.R.S.T.)
2 Faculdade de Engenharia, Universidade do Porto/s/n, R. Dr. Roberto Frias, 4200-465 Porto, Portugal
3 IDMEC, Instituto Superior Técnico (IST), Av. Rovisco Pais 1, 1049-001 Lisboa, Portugal; beatriz.silva@tecnico.ulisboa.pt (M.B.S.); fatima.vaz@ist.utl.pt (M.F.V.)
* Correspondence: jgil@inegi.up.pt

**Abstract:** Metallic additive manufacturing processes have been significantly developed since their inception with modern systems capable of manufacturing components for structural applications. However, successful processing through these methods requires extensive experimentation before optimised parameters can be found. In laser-based processes, such as direct energy deposition, it is common for single track beads to be deposited and subjected to analysis, yielding information on how the input parameters influence characteristics such as the output's adhesion to the substrate. These characteristics are often determined using specialised software, from images obtained by cross-section cutting the line beads. The proposed approach was based on a Python algorithm, using the scikit-image library and optical microscopy imaging from produced 18Ni300 Maraging steel on H13 tool steel, and it computes the relevant properties of DED-produced line beads, such as the track height, width, penetration, wettability angles, cross-section areas above and below the substrate and dilution proportion. 18Ni300 Maraging steel depositions were optimised with a laser power of 1550 W, feeding rate of 12 g min$^{-1}$, scanning speed of 12 mm s$^{-1}$, shielding gas flow rate of 25 L min$^{-1}$ and carrier gas flow rate of 4 L min$^{-1}$ for a laser spot diameter of 2.1 mm. Out of the cross-sectioned beads, their respective height, width and penetration were calculated with 2.71%, 4.01% and 9.35% errors; the dilution proportion was computed with 14.15% error, the area above the substrate with 5.27% error and the area below the substrate with 17.93% error. The average computational time for the processing of one image was 12.7 s. The developed approach was purely segmentational and could potentially benefit from machine-learning implementations.

**Keywords:** additive manufacturing; direct energy deposition; parameter acquisition; image single bead analysis

## 1. Introduction

Metallic additive manufacturing (MAM) is a family of technological processes where metallic components are manufactured by the successive deposition of cross-sections (or layers) of the final part until it is built [1]. This classification encompasses a considerable variety of processes, varying in complexity, productivity and cost; the present study focuses on direct energy deposition (DED), a process where the feedstock, which may be powdered (DED) or a wire (wDED) metallic compound, is melted by a heat source as it is being inserted into the melting pool [2].

The heat-source is usually a solid-state laser or a fibre laser [3], although an electron beam may be employed. In order to prevent corrosion phenomena, a localised inert atmosphere is created by the coaxial ejection of an inert gas—typically argon.

DED technologies may be compared to selective laser melting (SLM), another well-established MAM process, which produces parts through the successive melting of evenly spread metallic powder layers: at each cross-section, a thin coat (usually around 30 µm is spread by a powder roller or rake) before the heat source melts the powder as it is diverted through a system of mirrors to follow the cross-section of the built part. While DED features increased productivity at the cost of decreased part accuracy and surface finishing [4], often being named a near-net shape process [5], SLM is capable of producing more complex geometries, as it not only has better dimensional fidelity but also is capable of producing support structures [6].

Due to the aforementioned characteristics, SLM is capable of producing structural components with geometries of extreme complexity, such as implicit surfaces or internal porosities demanded by the aerospace [7] and biomedical [8] industries. However, due to the absence of a powder bed, DED is capable of depositing in existing surfaces with complex shapes, granting it the flexibility required for repairing applications [9] and surface cladding [10].

DED processes feature a considerable amount of processing variables, including the laser power, powder feeding rate, laser scanning speed, carrier gas flow rate, shielding gas flow rate, nozzle angle and nozzle-to-substrate distance, among others [2]. Furthermore, complex parameters may be analysed: researchers [11] have indicated a total of fourteen relevant dimensionless numbers that may be obtained to ascertain the deposition quality, efficiency and productivity; additional, non-dimensionless variables may be computed from the process parameters [2], translating them into concepts with physical significance that aids the development of optimised process parameter windows. The current work mentions the energy density $E$,

$$E = \frac{P}{v_s \times d_s} \tag{1}$$

in which $P$ is the laser power, $v_s$ is the scanning speed and $d_s$ is the laser spot size. The successful production of components through this technology thus revolves around the optimisation of the aforementioned variables for each material–substrate combination and its subsequent analysis, a process common for both DED and SLM technologies that is often expensive and time-consuming as it is based on trial-and-error depositions [2].

DED has already been used in depositing nickel-based super alloys, such as Inconel 625 [12,13] and Inconel 718 [14], 316L stainless steel [15], tool steels, such as AISI H13 [16], as well as in developing functionally graded materials (FGMs), which are components whose chemical and/or conditions show a spacial variation across the component [17]; SLM, on the other hand, has also been used to successfully deposit precipitation-hardened steel, such as 17-4PH [18], Ti6Al4V [19], 316L stainless steel [20], nickel super alloys [21] and aluminium alloys [22], among others.

A frequently used alloy within the context of MAM is Maraging steel, a ferrous alloy whose name is derived from the junction of the terms martensite and ageing as it obtains its mechanical strength from the precipitation hardening phenomenon resulting from ageing heat treatments [23]. Maraging alloys have very low levels of C as it contributes to the formation of fragile carbides, such as TiC [24]. Furthermore, it contains appreciable quantities of Ti, Al and Co and large quantities (between 18 and 25 weight %) of Ni, with the latter leading to a soft but heavily dislocated martensite [25] upon quenching.

These alloys find application in the automotive, aerospace, military and tool die industries [26]. One of the advantages of this alloy, apart from its exceptional combination of high strength and toughness, is its printability: a recent research work [27] conducted 16 single tracks in which the energy density (expressed in Equation (1)) oscillated from 66.7 J mm$^{-2}$ and 416.7 J mm$^{-2}$, returning beads with dilution proportions between 55% and 75%, values generally considered too excessive [12]; Additionally, researchers [28] developed processing maps for 18Ni300, concluding that laser powers larger than 1200 W with a laser spot size of 2.55 mm resulted in porosities below 0.7%, although speeds below

$9\,\mathrm{mm\,s^{-1}}$ for the same power setting lead to dilution proportions (given by Equation (2)) below 12.5%.

Recent work [29], in which 18Ni300 tensile specimens were obtained from nine different parameter combinations, concluded that energy density values over $180\,\mathrm{J\,mm^{-2}}$ lead to components with near 100% density, with the laser power being considered the most influential parameter on tensile property. Investigations into the influence of the powder feeding rate in depositing precipitation hardened steel on 304L stainless steel substrates were also conducted [30], where the dilution proportion was larger for increased feeding rates.

DED-related processes have different levels of in-situ control: recent research monitored the melting pool geometry during wire arc additive manufacturing to control for the bead's geometry and symmetry [31]; additional research was conducted on the use of a structured light system in the development of a real-time monitoring tool to aid the repair of an engine component [32]. Moreover, research towards establishing process maps that guide the deposition process towards avoiding lack-of-fusion and keyhole porosities has been conducted [33].

However, the aforementioned research mainly focuses on in-situ measurements and control over process variables, rather than output analysis, which is the case of this research work: the workflow associated with the parameter optimisation for different materials is similar, where different combinations of inputs are used in depositing single track line beads, which are subsequently cross-sectionally cut and analysed through optical microscopy (OM). In addition to probing for defects inherent to this technology, such as entrapped gas, lack-of-fusion pores and cracking [2], several macroscopic parameters were obtained, such as the track width $w$, height $h$, penetration $p$, wettability angles $\theta_l$ and $\theta_r$ and dilution proportion $D_p$; the latter is computed as:

$$D_p = \frac{A_b}{A_a + A_b} \cdot 100 \quad [\%],\tag{2}$$

where $A_a$ and $A_b$ are the areas above and below the substrate line, respectively. A schematic representation of these outputs is shown in Figure 1.

The computation of the aforementioned bead properties is usually done manually through image processing software, such as ImageJ; the novelty of this work thus arises in the development of an algorithm, using the scikit-image [34] package for Python, to obtain properties in a reliable and automated way. The OM images used as samples for developing and testing the developed algorithm were self-obtained, pertaining to the deposition optimisation of 18Ni300 Maraging steel on H13 tool steel substrates. It is important to mention that despite the images used in the benchmark of this code being comprised of 18Ni300 deposited on H13, this code should work regardless of the material as it is based on image segmentation and not in metallurgical expectations.

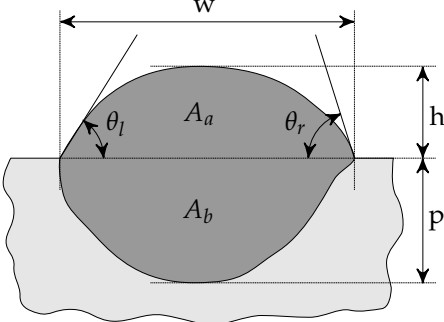

**Figure 1.** Schematic representation of DED-produced line bead properties.

## 2. Materials and Methods

### 2.1. DED Setup

The DED system where the samples were produced relies on a Coherent Highlight FL3000 fibre laser with a maximum power of 3000 W in continuous wave (CW) mode. Its wavelength is 1070 ± 10 nm with a focal diameter of 2.1 mm. The feedstock storage consists of two Medicoat AG disk powder feeders, delivering the powder to a Fraunhoffer powder splitter that subsequently diverts the powder flow towards four orifices in a COAX12V6 nozzle head, whose cooling system is capable of handling laser powers up to 6000 W. The employed inert gas is argon, fed by a cannister controlled by a pressure regulating valve set at 6 bar.

### 2.2. Depositions

#### 2.2.1. Powder and Substrate

The powder used in the depositions consists of Böhler W722 (Böhler Edelstahl GMBH, Kapfenberg, Austria), a grade equivalent to 18Ni300 Maraging steel, produced by inert gas atomisation [35]. Its morphology was analysed through scanning electron microscopy (SEM) with a FEI Quanta 400 FEG (FEI Company, Hillsboro, OR, USA), while its size distribution was analysed through dynamic light scattering (DLS) in a Coulter LS300 (Beckmen Coulter, Brea, CA, USA), by first suspending a sample of powder in ethanol and subjecting the solution to ultrassonic vibrations for one minute; results are shown in Figure 2, respectively. The powder's calculated $D_{10}$, $D_{50}$ and $D_{90}$ are 48.8, 89.8 and 143.7 µm, respectively.

The substrate was composed of Uddeholm Orvar 2M (FRamada, Aveiro, Portugal), an equivalent grade to AISI H13 tool steel, in its annealed state, at a temperature of 850 °C. The chemical composition of the substrate was obtained through mass spectroscopy with a SPECTROMAXx (Spectro, Berwyn, PA, USA) machine and whose results are shown in Table 1. The nominal values for this alloy were obtained from the supplier's website [36].

**Table 1.** Chemicals composition of the substrate, an AISI H13 tool steel alloy.

| H13 | Cr | Mo | Mn | P | C | Si | V | Fe |
|---|---|---|---|---|---|---|---|---|
| | | | **Chemical Composition [%]** | | | | | |
| Nominal | 4.80–5.50 | 1.20–1.50 | 0.25–0.50 | ≤0.03 | 0.35–0.42 | 0.80–1.20 | 0.85–1.15 | Bal. |
| Measured | 5.23 | 1.12 | 0.37 | 0.01 | 0.40 | 0.87 | 0.74 | Bal. |

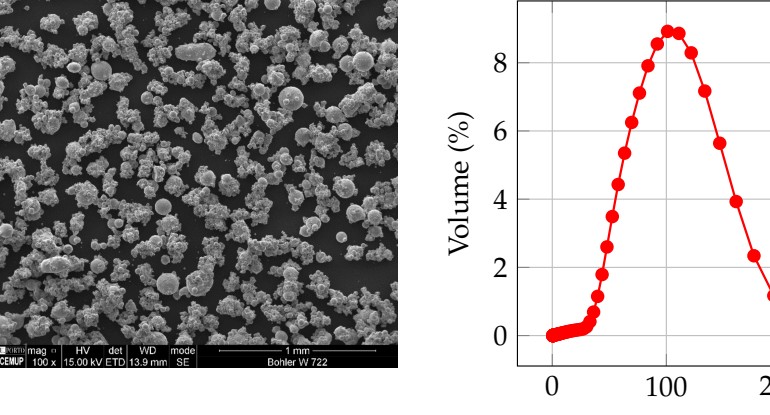

**Figure 2.** Powder morphology obtained through SEM and its size distribution calculated through DLS.

2.2.2. Process Parameters' Optimisation

The conducted depositions consisted on 25 mm-long 18Ni300 Maraging steel (W722 powder, Böhler, Germany) track beads on an AISI H13 (FRamada, Portugal) substrate, previously brushed and cleaned with acetone. The employed process parameters are shown in Table 2, with the parameter optimisation process consisting on the following procedure:

- Ideal outputs established according to three characteristics: the ideal dilution values were set between 10% and 30% [12,37], as very low percentages lead to detached lines while larger values may result in worse claddings [38]; optimised wettability angles were established to be between 50 and 70°, as decreased wettability angles are associated with increased oxidation [3] and increased angles worsen overlapping beads while depositing three dimensional objects [38]; parameters that lead to the reduction of defects inherent to DED, such as pores, cracking and keyhole porosities.
- Literature review: reviewed articles [27,28,39] suggest depositing lines, planes and/or specimens while maintaining energy densities above 65 J mm$^{-2}$, motivating an initial iteration orthogonal experiment L9 (3 × 3 Taguchi array) between the three most relevant parameters, namely the laser power $P$, scanning speed $v_s$ and feeding rate $f_r$. This DOE varied the energy density between $66.1 \leq E \leq 238.1$ J mm$^{-2}$ and the ratio between the scanning speed and feeding rate by $0.012 \leq v_s/f_r \leq 0.1$ m g$^{-1}$.
- Three additional L9 orthogonal arrays were deposited, whose parameter amplitude is expanded in Table 3.
- The shielding gas flow, carrier gas flow and distance to substrate were varied following a L9 orthogonal array, for the same laser speed, feeding rate and scanning speed that had generated the best outputs, according to the established metrics of wettability, dilution proportion and defects.

**Table 2.** The used parameters in the single track depositions pertaining to the parametrisation of 18Ni300 Maraging steel on H13 substrates.

| Track | Input Parameters | | | | | | |
|---|---|---|---|---|---|---|---|
| | $P$ [W] | $v_s$ [mm s$^{-1}$] | $f_r$ [g min$^{-1}$] | $q_s$ [L min$^{-1}$] | $q_c$ [L min$^{-1}$] | $d_n$ [mm] | $d_s$ [mm] |
| 1 | 1250 | 3.0 | 5.0 | 25 | 4 | 11.5 | 2.1 |
| 2 | 1250 | 9.0 | 5.0 | 25 | 4 | 11.5 | 2.1 |
| 6 | 1500 | 9.0 | 10.0 | 25 | 4 | 11.5 | 2.1 |
| 7 | 1500 | 3.0 | 15.0 | 25 | 4 | 11.5 | 2.1 |
| 10 | 1750 | 6.0 | 15.0 | 25 | 4 | 11.5 | 2.1 |
| 12 | 1250 | 3.0 | 5.0 | 25 | 4 | 11.5 | 2.1 |
| 16 | 1400 | 6.0 | 12.0 | 25 | 4 | 11.5 | 2.1 |
| 32 | 1550 | 3.5 | 12.0 | 25 | 4 | 11.5 | 2.1 |
| 33 | 1550 | 14.0 | 12.0 | 25 | 4 | 11.5 | 2.1 |
| 35 | 1400 | 12.0 | 12.0 | 30 | 4 | 11.5 | 2.1 |
| 38 | 1250 | 3.0 | 7.5 | 25 | 4 | 11.5 | 2.1 |
| 41 | 1250 | 2.0 | 5.0 | 25 | 4 | 11.5 | 2.1 |
| 50 | 1550 | 12.0 | 12.0 | 20 | 4 | 12.0 | 2.1 |
| 56 | 1550 | 12.0 | 12.0 | 30 | 4 | 13.0 | 2.1 |

It is important to mention that not all depositions were cross-sectionally cut, with the selection being guided by a visual inspection of the beads, along their respective parameters (analysis of lines with very similar parameters were avoided at an initial stage), as a way to save time and resources.

**Table 3.** Five DOEs conducted during the 18Ni300 process parameter optimisation. The $\pm$ sign indicates the variation of a parameter in one DOE: for example, the first DOE consisted of line beads with laser powers of 1250, 1500 and 1750 W, scanning speeds of 3, 6 and 9 mm s$^{-1}$ and feeding rates of 5, 10 and 15 g/ min.

| Parameter | DOEs | | | | |
|---|---|---|---|---|---|
| | 1st | 2nd | 3rd | 4th | 5th |
| Laser Power (W) | 1500 | 1400 | 1400 | 1400 | 1550 |
| $\pm$ | 250 | 150 | 150 | 150 | - |
| Scanning Speed (mm s$^{-1}$) | 6 | 6 | 9 | 3 | 12 |
| $\pm$ | 3 | 3 | 3 | 1 | - |
| Feeding rate (g min$^{-1}$) | 10 | 10 | 7.5 | 7.5 | 12 |
| $\pm$ | 5 | 5 | 5 | 5 | - |
| Shielding gas (L min$^{-1}$) | 25 | 25 | 25 | 25 | 25 |
| $\pm$ | - | - | - | - | 5 |
| Carrier gas (L min$^{-1}$) | 4 | 4 | 4 | 4 | 4 |
| $\pm$ | - | - | - | - | 1 |
| Nozzle distance (mm) | 11.5 | 11.5 | 11.5 | 11.5 | 12 |
| $\pm$ | - | - | - | - | 1 |

### 2.2.3. Image Acquisition

After concluding the depositions, the samples were cross-sectioned by a Remet TR-60 rotating disk (Remet, Italy), polished on Struers Rotopol-21 (Struers, Denmark) machines with SiC Buehler CarbiMet disks with 80, 180, 320 and 800 grits at 300 rpm for three minute each, followed by polishing with an alumina suspension on a Struers DP-U4 machine.

Lastly, the samples were polished with a diamond spray suspension of 6 μm for three minutes and 1 μm for three minutes, before being chemically etched with Nital 10% during 15 s. The images were acquired using a Leica DVM6 (Leica, Germany) digital stereoscope.

The line bead outputs were calculated using the open software Inkscape: the images were imported into the software's environment, and, using its drawing tools, different measurements were manually performed by either drawing on top of the bead for computing areas, or drawing single lines for the case of widths, penetrations and heights.

The wettability angles were found by establishing the horizontal limits of the bead and drawing lines tangent to the bead's contour near the interface with the substrate; the resulting lines' arc tangent returned the respective wettability angles. The measurement dimensions were recorded in pixels, and using the different image's scales, appropriate conversions between pixels and millimetres were performed.

### 2.3. Algorithm Workflow

Figure 3 schematically shows the workflow of the automated algorithm for image analysis. The operations associated with each step are detailed in the following section; the source code can be freely requested from the corresponding author, or downloaded as supplementary material in this article.

### 2.3.1. Reading Inputs

Inputs are requested in a `.txt` file named `criteria`; here, the user can control the scale factor from pixels to millimetres, give an indication on whether the deposited material is darker than the substrate's material, the fraction of the height considered when computing the wettability angles, whether or not visual outputs are to be saved, if a `.csv` file with the outputs is to be saved, the path to save data and lastly, the input image formats. The algorithm opens `.png`, `.jpeg` and `.tiff` files by default, and additional formats must be inserted in the file.

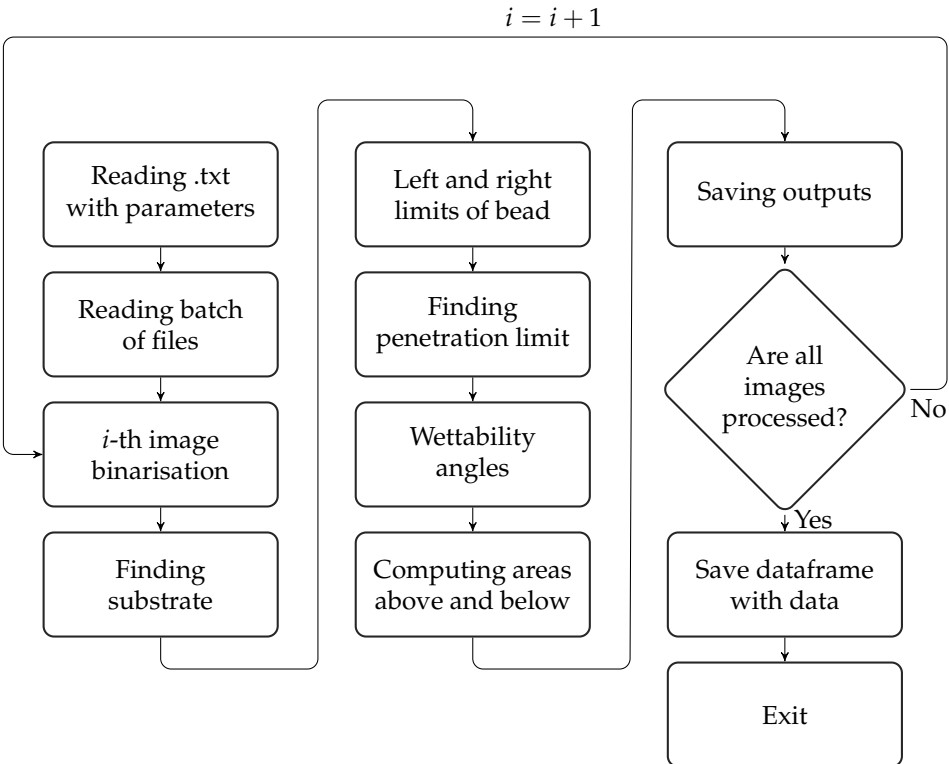

**Figure 3.** Developed algorithm's flowchart: user inputs limited to first step.

2.3.2. Sauvola Threshold and Binarisation

The Sauvola threshold is used to identify the pixels in the interface between the background and the substrate/line bead, as illustrated in Figure 4(2); this algorithm was implemented as it proved reliable in detecting interfaces on images with non-constant background lighting. The result of the Sauvola threshold is further de-noised through a white-tophat filter with a disk structuring element with a 1 px radius (Figure 4(3)). Larger radii in the structuring element may lead to discontinuities in the contour of the bead, which the algorithm "fixes" by connecting the discontinuities with a straight line, an approximation that should be avoided if possible.

The final binarised image is obtained by using the contours defined by the denoised Sauvola threshold and replacing the values in the background the value of 1 (one) and the coordinates in the foreground with the value of 0 (zero). The binarised image represents a binary version of the original image—Figure 4(4)—with the background possessing a pixel value of 1 (one) and both the substrate and line bead being 0 (zero) and is then used in detecting the substrate interface, as well as the thickness $w$ and height above the substrate $h$ of the bead, as explained in the following subsections.

2.3.3. Array Filtering

In the following subsections of this work, a filtering algorithm is mentioned, as it proved necessary to apply a low-pass filter in the computed arrays due to the oscillatory nature of pixel data, a problem that is further exacerbated in the subsequent spatial derivatives. The proposed solution uses a low-pass second order Butterworth digital filter with a sampling frequency of 500 Hz and a cutoff frequency of 0.04 Hz. Figure 5 shows the frequency response and cut-off frequency associated with the filter and the resulting filtered array.

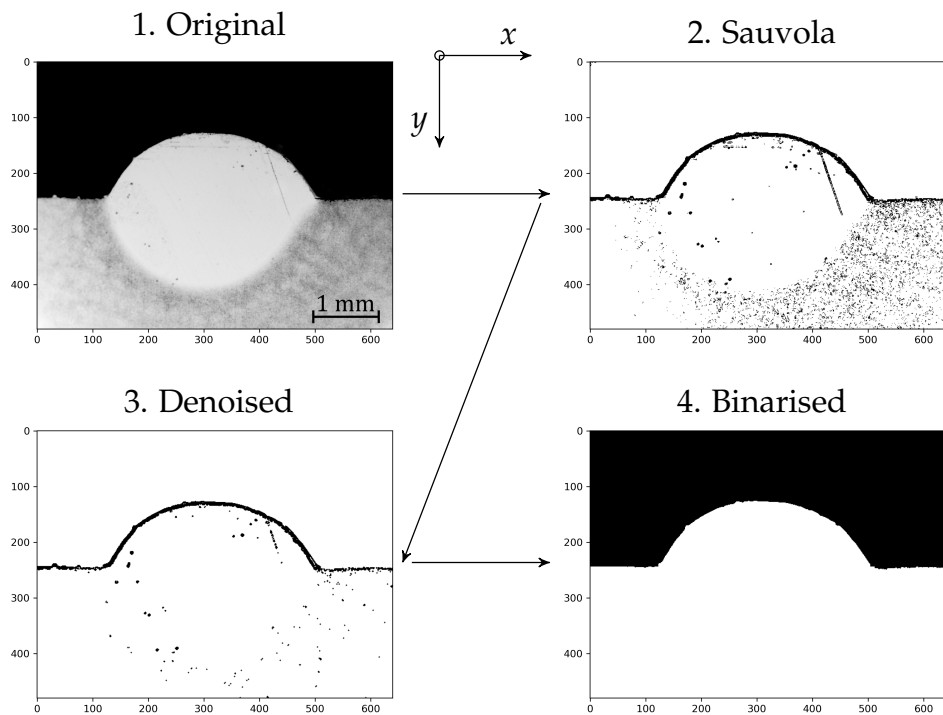

**Figure 4.** Stages associated with the binarisation of the line bead and the substrate (line bead displayed equates to Track 1). Step 1 is the original image; Step 2 is the result of applying a Sauvola threshold to the original image; Step 3 is the result of denoising the sauvola threshold with a white top-hat filter and a disk structuring element of radius 1 px; Step 4 is the binarised image, in which the background has value of 1 and the foreground has values of 0. The scale in the original image (Step 1) was added in post-processing and was not part of the image during the image processing.

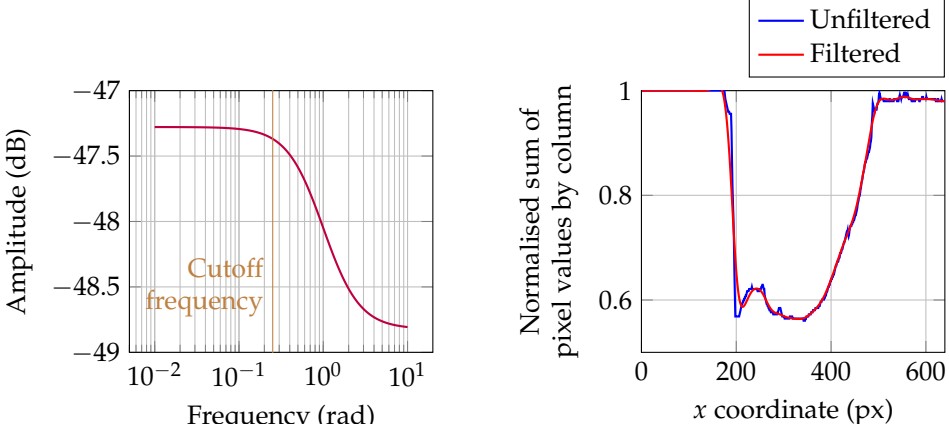

**Figure 5.** Frequency response pertaining to the sum of pixel values, by column, of the image belonging to track 33.

### 2.3.4. Substrate Identification

Identifying the substrate is crucial, as it establishes the division between the top and bottom portions of the line bead, hence, influencing the results of the bead height, penetration, wettability angles and dilution proportion. The proposed solution takes advantage of the sharp contrast between the substrate and the background's pixel values by computing the sum of all pixel values for each row in the binarised image of the line bead and subsequently normalising the obtained array.

This array is further differentiated with respect to $y$ in order to highlight two distinct coordinates at which significant gradients between pixel values occur: (i) the top of the

bead, which, in a denoised, binarised image, is the coordinate at which non-unitary pixel values first occur; (ii) the substrate line, which is the point at which unitary pixel values are replaced by zero-valued pixels in a denoised binarised image.

Figure 6 shows the placement of these two gradient peaks and relates to their afore-mentioned physical significance. Hence, the function `find_substrate_line_index` takes advantage of the global minimum of the derivative of the pixel value's normalised sum by row to take a first guess at the coordinate at which the substrate/line bead is located. Further checks are necessary, as certain beads display very low wettability angles and the global minimum may be situated at the top of the bead.

In this case, the function inspects the value of the normalised sum of the pixel rows at the coordinate at which the guess took place, minus 15% of the height of the analysed image. The idea behind this check is to eliminate false positives that are not the coordinate at which the substrate lies but rather another point. By checking the value of the normalised sum of pixel values by row, if indeed the guessed coordinate is the substrate interface, the expected value is near zero, as nearly all pixels on the binarised image should be null.

However, if the guessed coordinate is somewhere between the top of the analysed bead and the interface, a considerably fraction of pixels within that row will still hold unitary values, thus failing the check. The threshold that distinguishes the correct and incorrect guesses as established at 0.9.

The coordinate where the line bead reaches its maximum height is identified in a more straightforward manner, through function `get_line_bead_maximum`, which seeks the first occurrence along the $y$ rows of a zero-valued pixel.

### 2.3.5. Left and Right Limits

The left and right limits of the bead establish its width; a similar approach to that performed for finding the substrate index is employed, as the normalised sum of the binarised image pixel values is computed; although, in this situation, it is done for each column. The derivative of the filtered normalised sum was tried; however, it proved unreliable, computing a significant amount of false positives. A simpler approach was therefore implemented, in which a threshold value is established, the maximum height of the line bead is obtained, and the limits are defined as the interception between the sum of the pixel values by column with the threshold that lie closest to the bead's maximum. This method is depicted in Figure 7. The threshold is computed at one minus 5% the minimum normalised sum of the pixel values.

### 2.3.6. Penetration Limit Identification

The identification of the penetration limit, which is the coordinate at which the line bead most dilutes into the substrate, follows the logic of the substrate identification; how-ever, in this case, the sum of pixel values across each row must be performed on the grayscale image, as the binarised version eliminates the contrast between the line bead's dilution zone and the remaining substrate. This sum is only performed from the substrate line's index onward. The result is then filtered and differentiated, with the bottom of the melting pool to be associated with the maximum value of the derivative. Figure 8 illustrates how the maximum derivative relates to the penetration limit.

### 2.3.7. Wettability Angles

The wettability angle determination is done through the detection of false pixel values above the substrate line, in the binarised image of the bead. Each pixel coordinate at which the line bead is detected is appended into a list, which is then used to perform a linear regression, whose slope indicates the tangent of the wettability angle. There is some room for customisation in this function, as the height up to which pixels are considered depends on a user-defined criterion; the default value is a fraction of 0.4 of the line bead's height. Figure 9 highlights the influence of the height fraction when computing wettability angles.

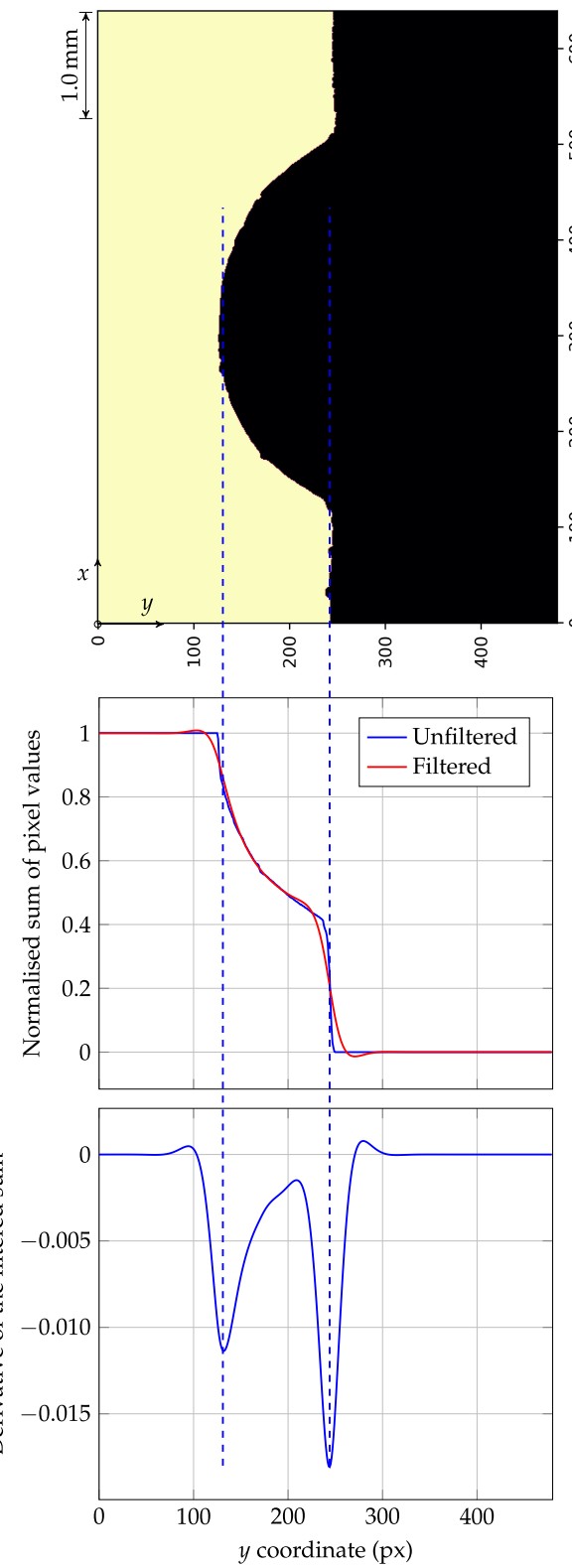

**Figure 6.** Operations associated with detecting the coordinate of the substrate's interface with the background, for line bead 1 (the first derivative's global minimum is one potential indicator of the substrate's starting location).

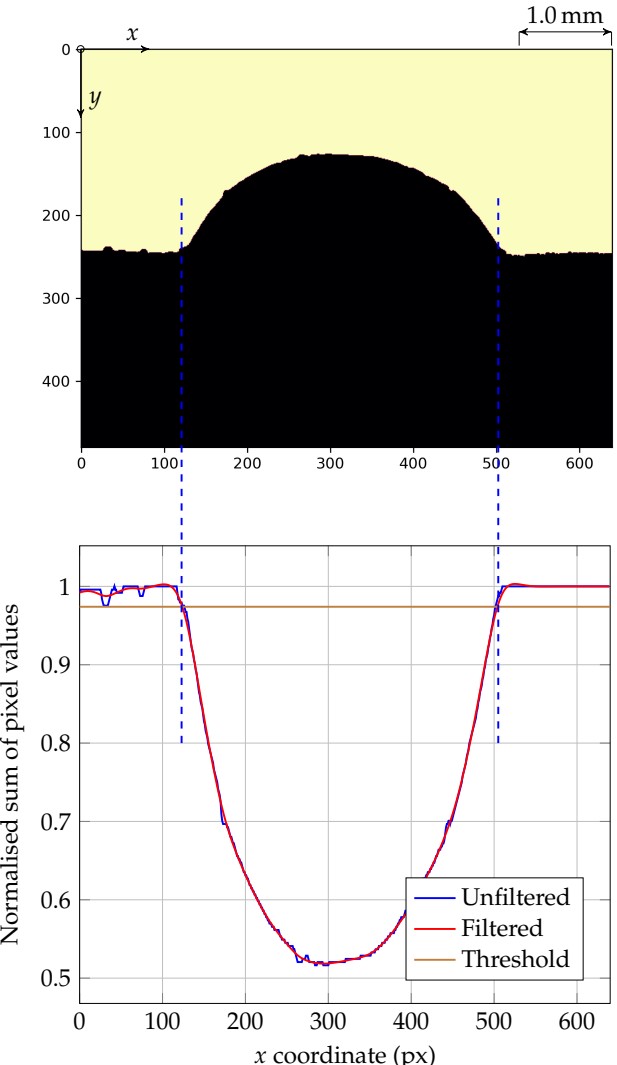

**Figure 7.** The detection of the bead's limits through the interception of the normalised sum of pixel values with an established threshold (shown track equates to track 1).

### 2.3.8. Areas

The computation of areas above and below the substrate line takes advantage of the image labelling functions already implemented in the `scikit` package. The sole hindrance to directly applying the label function is the segmentation of the diluted zone and its separation from the remaining substrate, as the input of the `label` function is a binary image. The isolation of the diluted zone is performed through the following steps:

1. The image is cropped to the diluted zone, which is delimited by the bead's thickness, the substrate line and the penetration limit.
2. The contrast of the image is adjusted via Gamma correction—`adjust_gamma` function—with a gamma value of 3.
3. The image is equalised—`equalize_hist` function.
4. The sum of all pixel values by column is computed;.
5. The resulting numerical values are mapped to the image, removing the pixel values whose coordinates lie below the mapped function.

The label function has several other properties built-in in addition to area computation and may be customisable according to the user's needs; the current version also calculates the bead's centroid, for instance. The calculated areas $A_a$ and $A_b$ are further used in

obtaining the dilution proportion, according to Equation (2). Figure 10 illustrates the several steps before the binarised dilution zone is obtained.

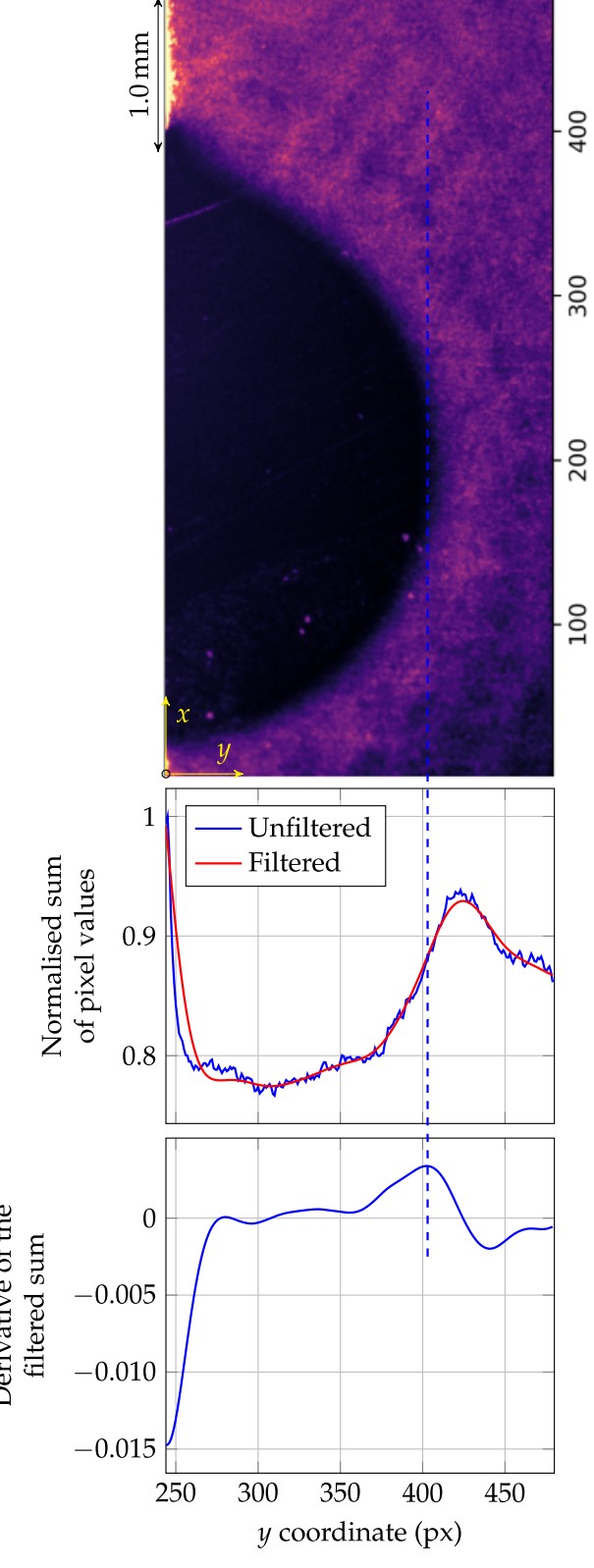

**Figure 8.** Determination of the penetration limit through the global maxima of the normalised sum of pixel values' first derivative.

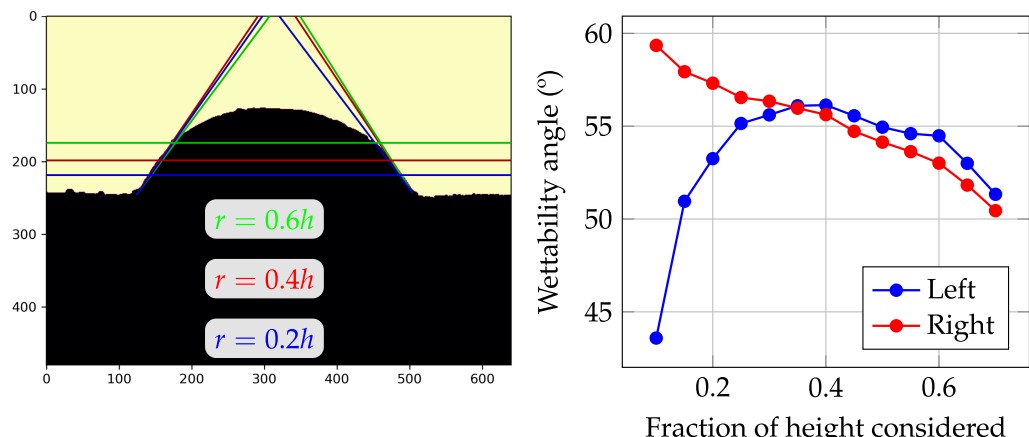

**Figure 9.** Influence of the height *r* considered when computing wettability angles (example shown pertains to track 1).

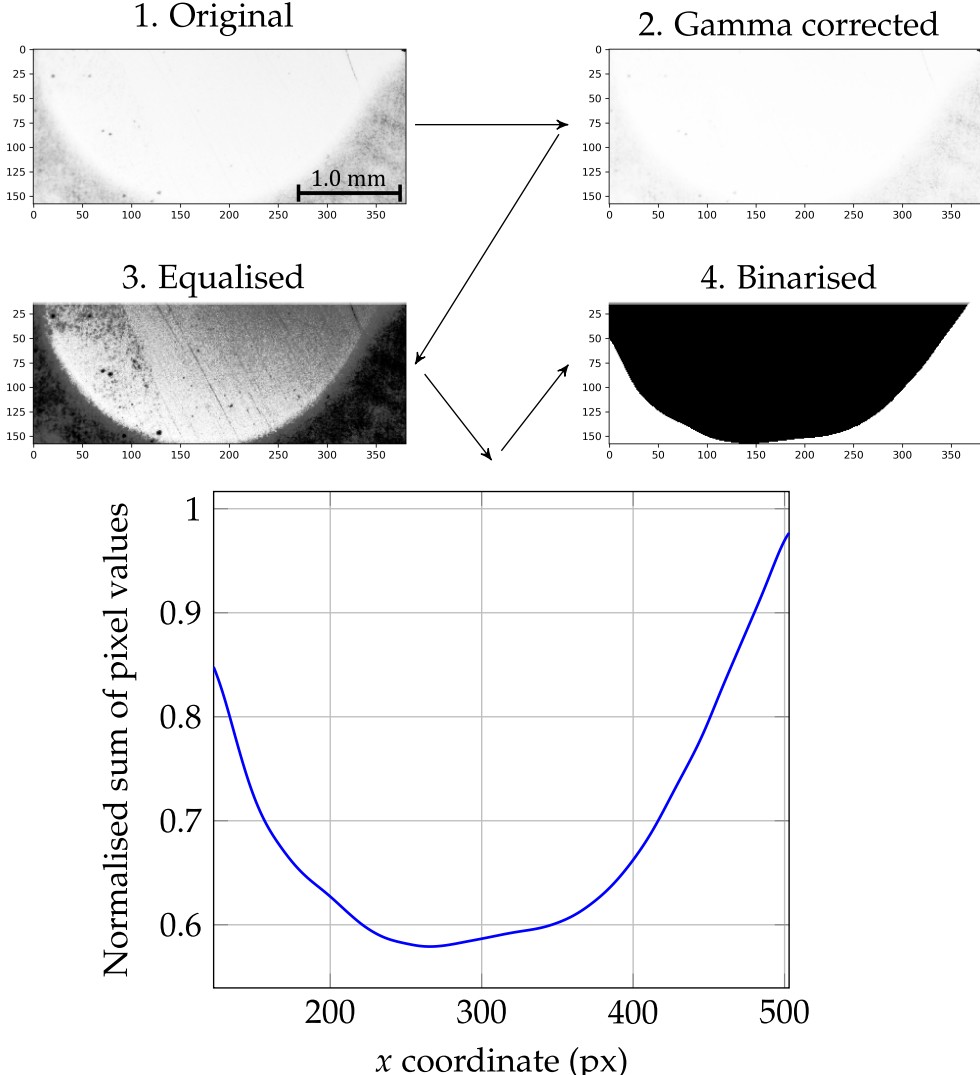

**Figure 10.** Stages associated with the segmentation of the diluted zone (example shown corresponds to track 1).

## 3. Results

Fourteen of the fifty-six line bead depositions were cross-sectioned and analysed through optical microscopy, obtaining images that served as inputs for the developed algorithm. Images of some of the cross-section images of the produced beads, overlaid in a plot between the laser power and laser spot size ratio $P/d_s$, and the ratio between the scanning speed and feeding rate $v_s/f_r$ are shown in Figure 11. The manually obtained results of the produced and analysed tracks are indicated in Table 4, while the algorithm-obtained results are indicated in Table 5. The error percentage between the values is included in Table 6. Samples of processed images are shown in Figure 12.

The parameter that displayed the best agreement between the manually obtained and numerically computed was the height above the substrate, $h$, while the worst parameter was shown to be the area below the substrate $A_b$. Furthermore, the width $w$, height $h$, penetration $p$ and area above the substrate $A_a$ displayed errors smaller than 10%.

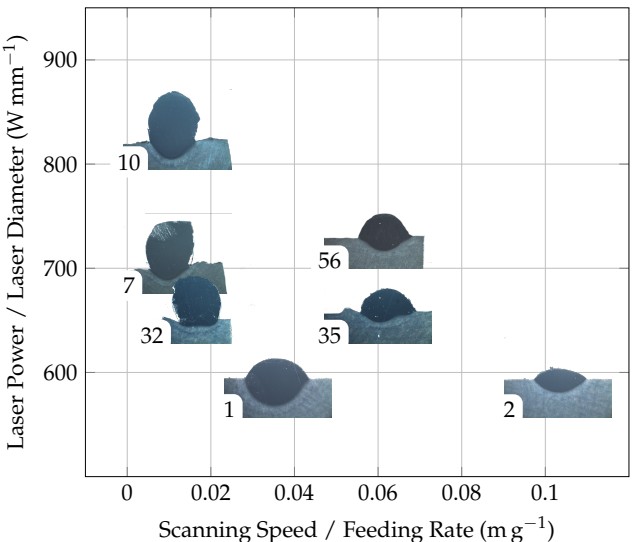

**Figure 11.** Parametrisation window overlaid with cross-section images of the produced beads: Lower ratios between the scanning speed and feeding rate lead to low dilution proportions and large wettability angles, while increased ratios lead to decreased bead heights.

**Table 4.** Output parameters determined manually.

| Track | Output Parameters | | | | | | | |
|---|---|---|---|---|---|---|---|---|
| | $w$ [mm] | $h$ [mm] | $p$ [mm] | $\theta_l$ [°] | $\theta_r$ [°] | $A_a$ [mm²] | $A_b$ [mm²] | $D_p$ [%] |
| 1 | 3.35 | 1.06 | 1.43 | 60.55 | 63.65 | 2.52 | 3.47 | 57.96 |
| 2 | 2.80 | 0.60 | 0.66 | 51.38 | 44.16 | 1.16 | 1.28 | 52.34 |
| 6 | 2.74 | 0.96 | 0.91 | 110.86 | 84.12 | 2.12 | 1.56 | 42.41 |
| 7 | 2.53 | 2.46 | 0.63 | 109.38 | 116.26 | 5.63 | 0.86 | 13.23 |
| 10 | 2.77 | 2.64 | 1.00 | 95.84 | 99.01 | 5.85 | 1.57 | 21.14 |
| 12 | 3.75 | 0.88 | 1.29 | 41.28 | 50.71 | 3.22 | 3.14 | 49.42 |
| 16 | 2.06 | 1.58 | 0.97 | 95.35 | 96.24 | 2.69 | 1.37 | 33.68 |
| 32 | 2.64 | 2.38 | 0.34 | 111.54 | 102.57 | 5.25 | 0.48 | 8.30 |
| 33 | 2.77 | 1.03 | 0.81 | 63.28 | 65.48 | 2.15 | 1.38 | 39.07 |
| 35 | 3.02 | 1.23 | 0.52 | 74.93 | 53.47 | 2.67 | 0.81 | 23.32 |
| 38 | 3.20 | 0.97 | 1.85 | 71.12 | 39.36 | 2.48 | 4.52 | 64.55 |
| 41 | 2.21 | 1.27 | 1.25 | 80.65 | 73.67 | 2.54 | 2.18 | 46.19 |
| 50 | 3.26 | 1.32 | 0.77 | 64.47 | 63.91 | 2.73 | 1.24 | 31.27 |
| 56 | 2.89 | 1.26 | 0.76 | 71.44 | 47.68 | 2.51 | 1.19 | 32.24 |

**Table 5.** Output parameters computed through the developed algorithm.

| Track | | | | Output Parameters | | | | |
|---|---|---|---|---|---|---|---|---|
| | $w$ [mm] | $h$ [mm] | $p$ [mm] | $\theta_l$ [°] | $\theta_r$ [°] | $A_a$ [mm$^2$] | $A_b$ [mm$^2$] | $D_p$ [%] |
| 1 | 3.38 | 1.06 | 1.43 | 56.86 | 56.17 | 2.64 | 3.57 | 57.51 |
| 2 | 2.72 | 0.66 | 0.64 | 49.28 | 38.01 | 1.16 | 1.28 | 52.56 |
| 6 | 2.84 | 1.03 | 0.89 | 74.44 | 62.88 | 2.10 | 1.70 | 44.79 |
| 7 | 2.62 | 2.43 | 0.69 | 100.28 | 102.6 | 5.61 | 1.32 | 19.03 |
| 10 | 2.81 | 2.54 | 1.04 | 98.36 | 104.22 | 5.67 | 2.06 | 26.65 |
| 12 | 3.57 | 0.94 | 1.20 | 49.44 | 61.68 | 2.54 | 2.90 | 53.35 |
| 16 | 2.13 | 1.57 | 0.98 | 84.74 | 83.13 | 2.78 | 1.58 | 36.24 |
| 32 | 2.63 | 2.51 | 0.17 | 114.43 | 167.87 | 5.73 | 0.29 | 4.74 |
| 33 | 2.78 | 0.98 | 0.80 | 66.90 | 56.99 | 2.14 | 1.53 | 41.68 |
| 35 | 3.05 | 1.28 | 0.32 | 74.25 | 52.98 | 2.77 | 0.71 | 20.32 |
| 38 | 3.27 | 0.96 | 1.86 | 64.20 | 37.01 | 2.16 | 4.98 | 69.69 |
| 41 | 2.29 | 1.33 | 1.32 | 103.45 | 71.24 | 2.46 | 2.40 | 49.43 |
| 50 | 3.04 | 1.23 | 0.78 | 56.83 | 44.59 | 2.62 | 1.72 | 39.60 |
| 56 | 2.96 | 1.29 | 0.70 | 66.00 | 47.66 | 2.65 | 1.30 | 32.87 |

**Table 6.** Percentage errors between the manually measured and algorithm-computed output parameters.

| Track | | | | Output Parameters' Errors | | | | |
|---|---|---|---|---|---|---|---|---|
| | $\epsilon_w$ [%] | $\epsilon_h$ [%] | $\epsilon_p$ [%] | $\epsilon_{\theta_l}$ [%] | $\epsilon_{\theta_r}$ [%] | $\epsilon_{A_a}$ [%] | $\epsilon_{A_b}$ [%] | $\epsilon_{D_p}$ [%] |
| 1 | 1.04 | 0.34 | 0.06 | 6.08 | 11.76 | 4.78 | 2.87 | 0.78 |
| 2 | 3.02 | 8.81 | 3.05 | 4.08 | 13.93 | 0.70 | 0.16 | 0.41 |
| 6 | 3.76 | 6.53 | 2.00 | 32.85 | 25.25 | 1.24 | 8.81 | 5.62 |
| 7 | 3.40 | 1.31 | 10.00 | 8.31 | 11.75 | 0.49 | 53.34 | 43.81 |
| 10 | 1.40 | 3.84 | 4.59 | 2.63 | 5.27 | 3.08 | 31.29 | 26.01 |
| 12 | 4.61 | 5.99 | 6.95 | 19.78 | 21.64 | 21.20 | 7.79 | 7.94 |
| 16 | 3.58 | 0.60 | 1.32 | 11.14 | 13.62 | 3.14 | 15.46 | 7.61 |
| 32 | 0.41 | 5.60 | 49.49 | 2.60 | 63.65 | 9.08 | 39.94 | 42.81 |
| 33 | 0.48 | 4.39 | 1.27 | 5.71 | 12.97 | 0.26 | 11.16 | 6.68 |
| 35 | 1.05 | 4.28 | 37.54 | 0.91 | 0.91 | 3.88 | 12.89 | 12.86 |
| 38 | 2.19 | 0.86 | 0.48 | 9.72 | 5.96 | 12.84 | 10.04 | 7.96 |
| 41 | 3.62 | 4.94 | 5.88 | 28.26 | 3.30 | 3.43 | 9.97 | 7.02 |
| 50 | 6.78 | 6.38 | 1.25 | 11.84 | 30.23 | 3.92 | 38.46 | 26.65 |
| 56 | 2.56 | 2.24 | 7.03 | 7.61 | 0.04 | 5.71 | 8.81 | 1.97 |
| Average | 2.71 | 4.01 | 9.35 | 10.82 | 15.73 | 5.27 | 17.93 | 14.15 |

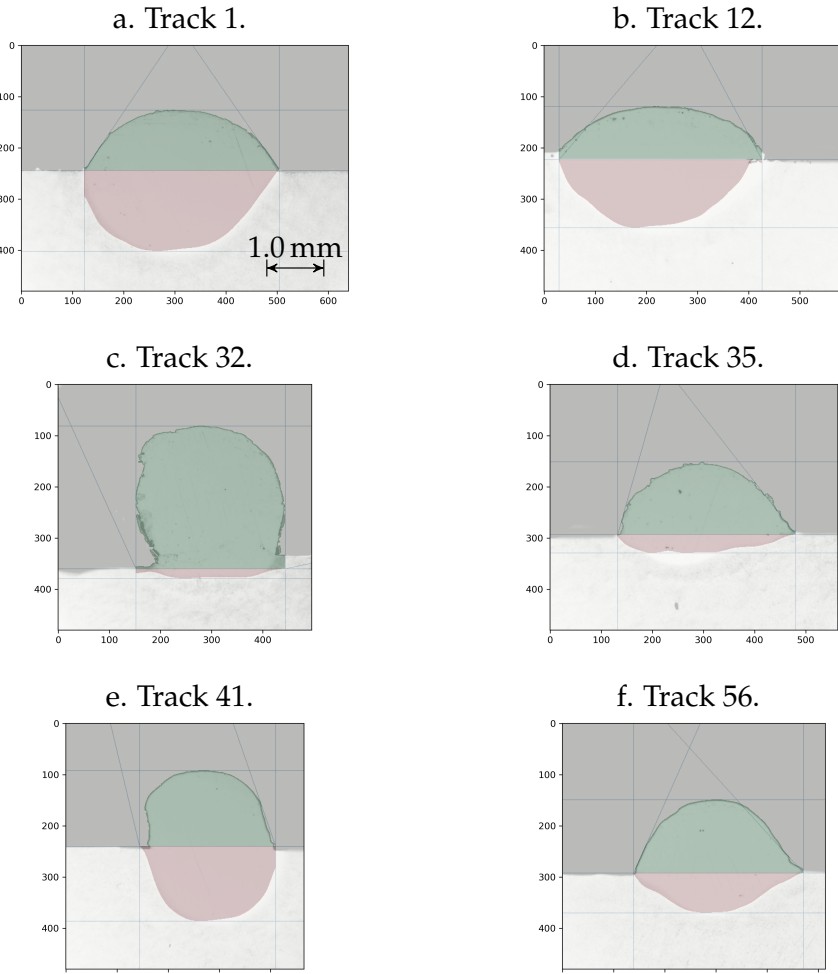

**Figure 12.** Examples of processed images with the elaborated algorithm (scale in pixels): the areas above and below the substrate are green and red, respectively; lines indicative of the width, height, substrate line, wettability angles and penetration limit are shown in blue. The images (**a**), (**b**), (**c**), (**d**), (**e**) and (**f**) are Tracks 1, 12, 32, 35, 41 an 56, respectively. Scale in the original image was added in post-processing and was not part of the image during the execution of the code.

## 4. Discussion

As stated in Section 2, the initial depositions followed a Taguchi L9 array between the scanning speed, feeding rate and laser power of the line beads; the parameters shown in Table 2 highlight four of the initial nine depositions, with their numbers being 1, 2, 6, 7 and 10, which were deemed the most interesting to further in additional DOEs: Track 1 displayed very large dilution values, attributed to its low relative scanning speed and low feeding rate. Track 2 also shows dilutions above the established limit of 30%, despite the larger scanning speed, attributing the dilution to the low feeding rate.

Track 6 improved the dilution proportion by decreasing it to 42.41%, although the wettability angles were too large and disparate. Tracks 7 and 10 also proved unsatisfactory, with wettability angles that were overly large. The following DOE oscillated process parameters around Track 1, by creating another L9 Taguchi array with laser power values of 1250 W, 1400 W and 1550 W, scanning speeds of $3\,\text{mm s}^{-1}$, $6\,\text{mm s}^{-1}$ and $9\,\text{mm s}^{-1}$ and feeding rates of $5\,\text{g min}^{-1}$, $9\,\text{g min}^{-1}$ and $12\,\text{g min}^{-1}$.

These depositions, of which Tracks 12 and 16 were analysed, yielded beads whose dilution proportion was too low and wettability angles too large, as easily perceptible through Figure 11, which shows the resulting outputs according to different parameters. The following depositions followed an orthogonal design and increased the scanning

speeds while reducing the feeding rates: the ratio between scanning speed feeding rate was restricted to an interval between 0.018 and 0.06 m g$^{-1}$.

Track 32 showed wettability angles that were too large, z possessed too large a dilution proportion, and finally Track 35 displayed acceptable values throughout. Further parameter variation ensued, with shielding gas flow rate, carrier gas flow rate, nozzle-to-substrate distance iterations resulted in beads with similar heights but considerably different dilution proportion and wettability angles; an increase in the disparity between the left and right angles in Track 56 could be attributed to the turbulence in the melting pool induced by the increase in shielding gas; Tracks 38 and 41 display large wettability angles, leading to the conclusion that very slow speeds ($\leq$3 mm s$^{-1}$) may increase the skewness of the produced bead.

Recent research [40] delved into the influence of the shielding gas flow rate on the porosity and quality of DED-produced 316L, concluding that the oxygen content present in the melt pool alters its surface tension, consequently affecting the melting pattern.

The feeding rate proved to have a significant impact on the dilution proportion, a conclusion also found in the literature, although the observed phenomena was contrary to the one reported by another research work [30], in which an increase in feeding rate led to larger dilution proportions. However, considering that the increase in powder delivered to the melt pool requires more heat energy to melt, it is expected for the available energy to be less capable of melting the substrate.

This conclusion is supported by similar research [41] that found the dilution proportion increases inversely when compared to the powder delivery per unit length. Additional research showed the possible instability due to excess power feed rate, which leads to poor overlapping [42]; track 7 and 10 highlight this problem, as their wettability angles are both greater than 90°.

The optimised process parameters resulted in a ratio between the laser power and laser spot size of $P/d_s =$720 W mm$^{-1}$ and a specific energy of $E =$61.51 J mm$^{-2}$; other research [29] deposited 18Ni300 tensile specimens using an orthogonal DOE between the laser power, scanning speed and feeding rate; the specimen produced with a specific energy of 67.7 J mm$^{-2}$ achieved the highest ultimate tensile strength. Additional research [28] optimised process parameters for 18Ni300 processing, obtaining specific energy of 41.95 J mm$^{-2}$ in the optimised parameters for dilution proportion. A comparison between process variables the present work and the aforementioned research is visualised in Figure 13.

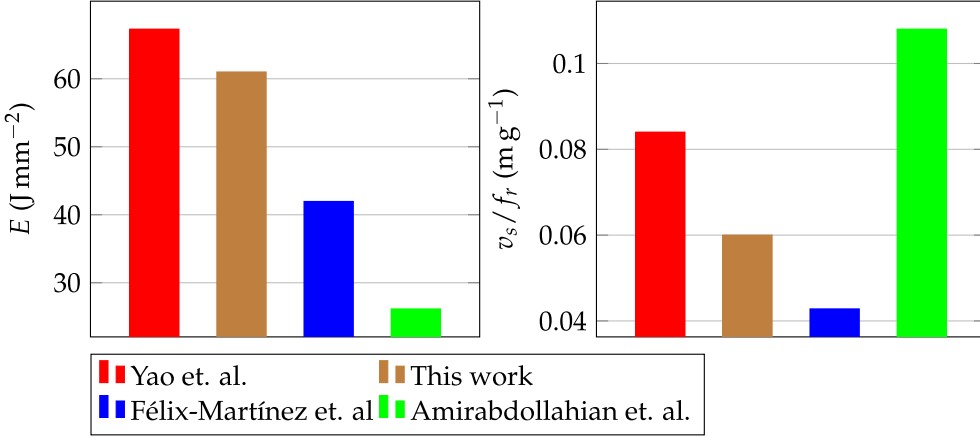

**Figure 13.** Comparison between the specific energy $E$ and ratio between scanning speed and feeding rate $v_s/f_r$ between the present work and other research: Yao et al. [29], Félix-Marínez et al. [28] and Amirabdollahian et al. [39]. The parameters shown by [28] were optimised for dilution $\geq$ 10%, and the work by Yao et al. [29] where the parameters resulted in a specimen with the largest tensile strength.

Regarding the developed algorithm and its output results, the width values proved to be the computed variable displaying the smallest error percentage, with the average error being 2.71%. This constitutes a satisfying error for the computed width, with no particular

outlier or track whose error is significant or would otherwise highlight a shortcoming in the algorithm. Regarding the height, track 2 shows a 8.81% error, attributable to a small chip developed during the cutting procedure of the sample that remained attached to the deposited part; this increase in height was not considered when manually measuring the height but was taken into account by the algorithm.

The penetration limit displayed two very evident outliers—track 32 and track 35—with both cases underestimating the values. In the former, the substrate line is significantly asymmetric—shown in Figure 12c, with the right and left sides of the track being misaligned by circa 50 px; as the computed substrate line sits near the substrate side, which is located nearer to the penetration limit, its value is underestimated when compared to the manually determined one. This measurement could be improved by considering the differential between the substrate height to the left and to the right hand sides of the bead.

Alternatively, track 35—Figure 12d—miscalculated the penetration limit; the dilution zone of this track is asymmetric and presents a bulge to the left side of the dilution zone, which the algorithm failed to include, thus underestimating the penetration limit and the area below the substrate. This could be avoided by additional criteria in detecting the penetration limit in addition to the maximum of the derivative of the filtered sum of pixel values by row, similarly to the additional criteria used when detecting the substrate.

The wettability angles displayed errors of 10.82 and 15.73% for the left and right angles, respectively. Interestingly, it proved common for analysis to return angles with significant error disparity, such as tracks 12, 16, 33, 41 and 50; a potential explanation is the fact that both angles are measured considering the substrate line, instead of taking into account a possible height difference between the left and right hand side of the beads. The area below the substrate $A_b$ should also be improved, with tracks 7, 10, 32 and 50 displaying errors above 30%; in these cases, the area is considerably overestimated due to poor mapping of the sum of pixel values by column into the melting pool dimensions, stretching the area to the horizontal limits of the line bead, instead of the horizontal limits of the diluted zone.

In short, the current function works under the assumption that the area under the substrate's width and the line bead's widths are the same, a consideration that should be avoided in future adaptations. The larger error in the computation of the area below the substrate is ultimately due to the lesser contrast between the substrate and the diluted zone in the bead, when compared to the contrast between the background of the image and the line itself, for example.

## 5. Conclusions

In this work, we suggested an approach to automate the computation of properties from DED-produced tracks using Python and the `scikit` package. The developed algorithm was tested using self-produced optical microscopy imaging from 18Ni300 Maraging steel DED depositions on H13 tool steel substrates using different parameters until optimised inputs were achieved. The approach resulted in the following assertions:

- Track beads were successfully produced and analysed with a minimal presence of pores and defects: the optimised parameters consisted of a ratio between the laser power $P$ and laser spot diameter $d_s$ of $750\,W\,mm^{-1}$, a ratio between the scanning speed and feeding rate of $0.06\,m\,g^{-1}$, a shielding gas flow of either 25 or $30\,L\,min^{-1}$, depending on whether the application was cladding-based or component-based, carrier gas flow rate of $4\,L\,min^{-1}$ and a laser stand-off distance of 12 mm.
- A functioning tool capable of analysing single track depositions by identifying the substrate, areas above and below the substrate, the height, width, penetration and wettability angles were successfully built with an average analysis time per bead of 12.7 s. While this amount of time is considerable, it is worth mentioning that this is an improvement on the time it would take a human being to perform equal tasks, especially considering a large batch of images, such as presented in this work.
- Width and height properties were ascertained with errors of 2.71 and 4.01%, respectively.

- The area above the substrate and area below the substrate were computed, displaying errors of 5.27 and 17.93%, respectively. The increase in error of the area below the substrate when compared to remaining outputs lies in the less pronounced contrast between the diluted zone and the substrate when compared to the contrast between the foreground and the background of the image.
- Dilution proportion values were computed with an average error of 14.15%, and with line beads whose dilution zone was smaller displayed worse accuracy.

**Supplementary Materials:** The following supporting information can be downloaded at: https://www.mdpi.com/article/10.3390/app12052755/s1.

**Author Contributions:** Investigation, J.G.; Software, J.G., M.B.S., M.F.V., A.R.; Writing—original draft, J.G.; Conceptualization, A.d.J., J.M.R.S.T.; Methodology, M.B.S., M.F.V., A.R.; Funding Acquisition, A.d.J., A.R.; Writing—review and editing, A.d.J., J.M.R.S.T.; Supervision, J.M.R.S.T. All authors have read and agreed to the published version of the manuscript.

**Funding:** The authors would like to acknowledge the funding of the PhD scholarship with the reference UI/BD/150684/2021, funded by Fundação para a Ciência e Tecnologia. The authors acknowledge the funding through the Add.Strength project, with reference PTDC/EME-EME/31307/2017, funded by FEDER and FCT. The authors would like to thank the support provided by Fundação para a Ciência e a Tecnologia of Portugal and IDMEC under LAETA-UIDB/50022/2020.

**Institutional Review Board Statement:** Not applicable.

**Informed Consent Statement:** Not applicable.

**Data Availability Statement:** Data is made available and free to use through the supplementary data in MDPI's website.

**Conflicts of Interest:** The authors declare no conflict of interest.

## Abbreviations

The following abbreviations are used in this manuscript:

| | |
|---|---|
| MAM | Metallic additive manufacturing |
| DED | Direct energy deposition |
| wDED | Wire direct energy deposition |
| OM | Optical microscopy |
| FGMs | Functionally graded materials |
| DLS | Dynamic light scattering |
| DOE | Design of experiment |
| SEM | Scanning electron microscopy |

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
