# Peer review of "Automation of Property Acquisition of Single Track Depositions Manufactured through Direct Energy Deposition"

_applsci, doi:10.3390/app12052755_

Round 1

Reviewer 1 Report

This work suggested an approach to automate the computation of properties from DED-produced tracks, using Python and the scikit package. The developed algorithm was tested using self-produced optical stereoscopy imaging from18Ni300Maraging steel DED depositions on H13 tool steel substrates, using different parameters until optimized inputs were achieved. The paper is significant for the industrial application of 3D printing, but the following comments should be addressed.

  • The developed algorithm was tested using 18Ni300Maraging steel DED depositions on H13 tool steel substrates. Is the developed algorithm dependent on the used material? If the used material is changed (such as Ti alloys or Al alloys), whether the tested results will be same or not?
  • The development process on the algorithm is lack. The authors should provide more details about how to establish the developed algorithm.
  • The average analysis time per bead is 12.7 s. It is too long for the industrial application. How to shorten the analysis time?
  • Why is the average error of dilution proportion values obviously larger than the height, width and wettability angles?
  • The number of tested tracks (56 tracks) is too small. I suggested more tested tracks should be counted to improve the reliability.

Reviewer 2 Report

The present work proposes automation of property acquisition of single track depositions manufactured through direct energy deposition. The paper is well written (but the article needs a typo-grammatical check). Here are the main comments about the paper which need to be considered:

  • The introduction was performed correctly. However, it requires adjustments. It should provide a broader description of the research in other publications. 
  • Chapter 2 presented the correct description of the experiment. 
  • The discussion chapter needs to be expanded. The authors in this chapter refer only to 2 publications [21 and 22]. Therefore a more detailed literature review should be performed in order to be able to compile the reader how the research conducted in the article relates to the previously conducted research.

Reviewer 3 Report

Dear Authors! I have reviewed your manuscript "Automation of property acquisition of single track depositions manufactured through direct energy deposition". It is a sound report and I congratulate you a good piece of work. Please, answer my minor questions and comments before publishing. 

Reviewer 4 Report

  1. How was the DOE on Table2 was selected? Add more detail.
  2. The explanation of Figure 4 needs more improvement.
  3. Add a short note about the results to the abstract.
  4. Please add a scale bar to Figure 4.
  5. How authors selected the process parameters for the experimentation. This is important for repeating the test.
  6. The text has some typos. Please check them.
  7. The introduction needs to be updated by comparing the DED and Laser based powder bed fusion LB-PBF which is also called SLM. Read and add the following new references.
  • Evolution of temperature and residual stress behavior in selective laser melting of 316L stainless steel across a cooling channel
  • Fatigue life optimization for 17-4Ph steel produced by selective laser melting
  • Proposal of design rules for improving the accuracy of selective laser melting (SLM) manufacturing using benchmarks parts
  • Study of anisotropy through microscopy, internal friction and electrical resistivity measurements of Ti-6Al-4V samples fabricated by selective laser melting
  • High-cycle fatigue properties of curved-surface AlSi10Mg parts fabricated by powder bed fusion additive manufacturing
  1. Additive manufacturing has many advantages over the conventional manufacturing method which can be highlighted in your paper. Please read the following article and add to the introduction to show the experimental application of additive manufacturing and the advantage of this process over conventional manufacturing like machining.

Additive manufacturing a powerful tool for the aerospace industry.

Round 2

Reviewer 1 Report

I am satisfied with the response.

Reviewer 4 Report

The paper is ready to go.